

# Expression of 1B capsid protein of Foot-and-mouth disease virus (FMDV) using baculovirus expression system and its validation in detecting SAT 2- specific antisera

Wael Elmenofy[1], Ismail Mohamed[1], Lamiaa El-Gaied[1], Reda Salem[1], Gamal Osman[1,2,3] and Mohamed Ibrahim[4]

[1] Agricultural Genetic Engineering Research Institute (AGERI), ARC, Giza, Egypt
[2] Department of Biology, Umm Al-Qura University, Makkah, Makkah, Saudi Arabia
[3] Research Laboratories Center, Faculty of Applied Science, Umm Al-Qura University, Makkah, Saudi Arabia
[4] Department of Molecular and Cell Biology, University of Texas at Dallas, Richardson, Texas, USA

## ABSTRACT

Foot-and-mouth disease virus (FMDV) is one of the most devastating animal viruses that affect livestock worldwide. The 1B capsid of FMDV has been widely used to detect and confirm the infection. In the present study, the sequence coding for 1B subunit of FMDV capsid was expressed in insect cells using the baculovirus expression system under the polyhedrin (*polh*) promoter. The expression of 1B capsid protein was validated in the culture filtrate of insect cells using SDS-PAGE and western blotting. The culture filtrate containing recombinant 1B capsid (r1B) was used as a coated antigen in an indirect enzyme-linked immunosorbent assay (ELISA). The antigenicity and specificity of r1B against SAT 2 serotype-specific antibodies were assessed. Our results revealed that a protein concentration as low as 25 ng could detect SAT 2-specific antibodies in ELISA. The results highlight the application of insect cells developed r1B protein in the detection of FMDV. Further studies are required to determine the ability of r1B to detect other FMDV serotypes.

## INTRODUCTION

Foot-and-mouth disease (FMD) is considered as one of the most devastating diseases that affect the livestock production (*Kitching et al., 2007*; *Arzt et al., 2010*) and results in clinical signs such as fever, general weakness, and vesicular sores on the tongue, gag, feet, and nipples (*Grubman & Baxt, 2004*; *Arzt et al., 2011*). FMD is caused by aphthovirus, a viral genus from the family Picornaviridae. There are seven distinct serotypes namely, A, O, C, Asia 1, and South African Territories (SAT) 1, 2, and 3, representing FMDV (*Belsham, 1993*; *Salem et al., 2019*). These serotypes exhibit significant antigenic and hereditary assorted varieties, resulting in several subtypes and genealogies. Accordingly, animals may become

Corresponding author
Gamal Osman, geosman@uqu.edu.sa

resistant to one serotype but still show sensitivity to some other serotype (*Domingo et al., 1993*). Only serological tests could verify the presence of 60 FMDV. Serological FMDV tests are conducted for the following purposes: verify suspected cases, certify individual animals for trade before travel, demonstrate the absence of disease, and demonstrate vaccine effectiveness. FMDV serological tests are of two types: one that detects non- structural protein (NSP) antibodies and the other detects structural protein (SP) antibodies. The tests based on SP detect antibodies triggered by infection or vaccination (*Golding et al., 1976*; *Hamblin, Barnett & Hedger, 1986*; *Hamblin et al., 1987*; *Salem et al., 2020*). The SP-based tests are ideal for tracking the immunity conferred by field vaccination, verifying past or ongoing disease in non-vaccinated animals, certifying pre-transportation animals (for commercial purposes). The latest FMDV detection techniques use inactivated live virus that presents some health risks. Moreover, these are time-consuming, technically challenging, and have a significant rate of false-positive reactions. The structural proteins of FMDV are very complex and any alteration in their structures allows FMDV to escape from the host immune system, which complicates the detection of antibodies against each serotype (*Carrillo et al., 2005*). Therefore, having a sensitive, rapid, and reproducible laboratory test to detect FMDV antibodies irrespective of serotypes is crucial.

The 1B capsid is a relatively conserved region among FMDV serotypes; however, certain replacements in its amino acids have resulted in considerable antigenic diversity, suggesting that 1B capsid harbors important antigenic epitopes (*Acharya et al., 1989*). In the present study, we created and expressed a recombinant 1B capsid protein to be used as a coating antigen in a simple local indirect ELISA for detecting antibodies to FMDV, regardless of its serotype and as an alternative to inactivated FMD viral antigen to confirm previous or ongoing infection in non-vaccinated animals and to certify animals beforehand. Amino acid sequence alignments of these capsid proteins revealed that 1B capsid is the most highly conserved among several FMDV subtypes. We believe that 1B capsid could be used to detect FMDV subtypes (*Jamal & Belsham, 2013*; *Salem, El-Kholy & Ibrahim, 2019*). In the present study, the 1B capsid protein was expressed using the baculovirus/Sf9 cell expression system. The construct was designed to direct the secretion of the recombinant protein into cell culture media. We hypothesized that the secretion of recombinant protein could minimize the misfolding of nascent peptides, thus reducing the likelihood of the protein to aggregate. Furthermore, we believe that the expression in insect cells allows for post-translation modifications that could trigger a stronger immune response than that obtained from proteins expressed in a prokaryotic expression system.

## MATERIALS AND METHODS

### Cells, virus, and antibodies

The Sf9 cells were maintained at 27 °C in Ex-Cell 420 serum-free medium (Sigma) supplemented with antibiotic/antimycotic (Sigma). The Bac-to-bac baculovirus expression system vector was used to clone and express 1B capsid protein in Sf9 cells (Thermo Fisher Scientific). Antibodies used included monoclonal anti-His tag antibodies as the primary antibody and horseradish peroxidase (HRP)-coupled anti-bovine IgG (Promega) as the secondary antibody.

## Designing, synthesis, and cloning of the 1B capsid gene

FMDV-SAT 2 isolate (gb|AAZ83686) was used as the reference strain for cloning the 1B capsid gene. In brief, the polypeptide that spans the 1B capsid flanked by 8 amino acid residues at the N-terminus (the loop connecting 1A and 1B), and 30 amino acid residues at the C-terminus (loop connecting 1B and 1C) was selected for expression. The corresponding nucleotide sequence that codes for 1B polypeptide was subjected to codon usage optimization so that it better matched *Spodoptera frugiperda*, the surrogate cells used for expression. The coding sequence of six histidine residues (Histidine tag) and an enterokinase recognition sequence were also introduced at the 3′ end of the 1B expression cassette. Collectively, this polypeptide consists of 200 amino acids. The 1B capsid expression cassette was cloned into the BssHII/PstI sites of the pFastBac cloning vector and the expression of the whole cassette was driven by the *Ppol* promoter.

## Generation of recombinant bacmid in *Escherichia coli*

In order to generate recombinant bacmid that harbors the 1B capsid gene, the recombinant 1B/pFastBac$^{TM}$ construct was transformed into *Escherichia coli* strain DH10Bac as per the manufacturer's instructions (Life Technologies). Briefly, approximately 1 ng of the 1B/pFastBac$^{TM}$ DNA in a total volume of 5 μL was transformed into 100 μL of pre-chilled competent DH10Bac cells. The mixture was kept on ice for 30 min and heat-shocked for 45 s at 42 °C. Immediately, the transformed cells were chilled on ice for 2 min. Around 900 μL of the Luria-Bertani broth (LB) medium was added and the bacterial cells were incubated at 37° C with shaking for 4 h. This incubation period was sufficient to facilitate transposition of the recombinant cassette into the bacmid mini-*att* Tn7 site within the *lac*Z α-complementation region of the bacmid located in DH10Bac genome. Five serial dilutions ($10^{-1}$, $10^{-2}$, $10^{-3}$, $10^{-4}$, and $10^{-5}$) of the obtained culture were prepared and spread onto the LB agar plates (50 μg kanamycin, 7 μg gentamicin, and 10 μg tetracycline). Furthermore, X-Gal (100 mg/mL) and isopropyl β-d-1-thiogalactopyranoside (IPTG) (40 μg/mL) were added to each plate to enable the screening of blue/white colonies. All plates were again incubated at 37 °C for 48 h.

## Verification of successful transposition of bacmid DNA

In order to verify successful transposition, 20 white colonies were selected for screening with PCR using oligos M13/F (5′-GTTTTCCCAGTCACGAC-3′) and M13/R (5′-CAGGAAACAGCTATGAC-3′). These oligonucleotides flanked the insertion site of mini-*att* Tn7 in the recombinant bacmid.

## Isolation of recombinant bacmid from *E. coli*

A polymerase chain reaction (PCR)-positive bacterial colony was picked and used to inoculate two mL LB medium supplemented with 50 μg/mL kanamycin, 7 μg/mL gentamicin, and 10 μg/mL tetracycline. The bacterial culture was incubated overnight at 37 °C with shaking. Bacterial cells were collected at $14,000 \times$ g for 1 min and suspended in 0.3 mL of Solution I (15 mM Tris-HCl, pH 8.0, 10 mM EDTA, and 100 μg/mL RNase A). Cells were gently mixed and incubated at room temperature (RT) for 5 min. Approximately, 0.3 mL of Solution II (0.2 N NaOH, 1% sodium dodecylsulfate [SDS]) was added, followed

by 0.3 mL of 3.0 M potassium acetate (pH 5.5). The bacterial lysate was chilled on ice for 10 min. The clear lysate, containing the bacmid DNA, was separated from the insoluble cell debris by centrifugation for 10 min at $14,000\times$ g. The supernatant was transferred in a new microfuge tube containing 0.8 mL of absolute isopropanol. The DNA pellet was collected by centrifugation for 15 min at $14,000\times$ g at RT and washed twice using 70% ethanol. The DNA pellet was air-dried at RT for a few minutes and dissolved in 50 µL Tris-EDTA (TE) buffer.

## Transfection and propagation of recombinant baculovirus in Sf9 cells

Recombinant bacmid DNA was used to transfect Sf9 cells according to the method described by *O'Reilly, Miller & Luckow (1992)*. Briefly, 500 ng bacmid DNA was mixed with Cellfection (Life Technologies). To this, 210 µL of Excell-420 serum-free medium was added and the solution was kept at RT for 30 min. This mixture was added dropwise to Sf9 cells ($5 \times 10^9$ cells/plate), previously grown in a 35-mm tissue culture six-well plate. The cells were incubated at 27 °C for 5 h. Subsequently, the medium containing Cellfection/bacmid DNA mixture was removed and replaced with Excell-420 serum-free medium, supplemented with the appropriate antibiotics. The cells were incubated at 27 °C for 72 h in a humidified incubator until the appearance of signs of viral infection. Successful transfection was confirmed by observing the cytopathic effect (CPE) due to viral infection under inverted light microscopy.

## Detection of recombinant virus in infected Sf9 cells by PCR

After the transfection of Sf9 cells with recombinant bacmid, it was necessary to verify the presence of viral DNA in the infected cells. Accordingly, PCR was used to detect viral-related amplicons in Sf9 cells. PCR reactions were performed using three primers specific for the 1B capsid gene: C1B/F (5-CCT AAC ACC TCA GGT CTG GAG ACT CG-3), C1B/R1 (5-GGT GTA CGA ATC GGT CAG CTT GC-3), and C1B/R2 (5-ATT GAT GAA CTG GTG AGG GTA GAG G-3). The expected amplicon sizes using C1B-F/C1B-R1 and C1B-F/C1B-R2 are 162 bp and 306 bp, respectively. The PCR reactions were performed in a total volume of 25 µL. The PCR reaction mixture was performed using the following contents: 10 pmol of each primer, 1 µL $MgCl_2$ (25 mM), 1.5 µL of dNTPs (10 mM each), 2 µL of DNA (0.1–0.5 µg), 5 µL of 5× PCR reaction buffer, and 0.5 µL of GoflexiTaq DNA polymerase (5 U/µL) (Promega). The PCR cycles were as follow: an initial denaturation cycle at 95 °C for 3 min, followed by a total of 35 cycles of denaturation at 95 °C for 1 min, annealing at 55 °C for 1 min, and extension at 72 ° C for 1 min; and final cycle at 72° C for 7 min to allow the completion of primer extension.

## Viral DNA isolation from infected Sf9 cells

The purpose of this experiment was to isolate the recombinant virus harboring the 1B capsid protein from Sf9 cells. To achieve this, four mL of the bacmid-infected Sf9 cells was spun down and the cells were collected and dissolved in 500 µL double distilled water ($ddH_2O$). Recombinant virions were released from Sf9-infected cells by incubation in 0.1 M $Na_2CO_3$ at 37 °C for 1 h. The solution was neutralized to pH 8.0 with few drops of 1.0 M HCl and subsequently treated with 45 µg/mL RNase A at 37 °C for 10 min. The solution

was incubated in 1% SDS and treated with 250 µg/mL Proteinase K for 1 h at 37 °C. The solution was extracted twice with Tris-EDTA (TE)-saturated phenol:chloroform:isoamyl alcohol mixture (25:24:1 [v/v/v]), followed by chloroform:isoamyl alcohol (24:1 [v/v]) extraction. The DNA was collected in 2.5 volume of ice-cold 97% ethanol in the presence of 1/10 volume 3.0 M NaOAc, pH 5.2, for 10 min at 14,000 rpm. The DNA pellet was washed twice with 70% ethanol, recollected, and dissolved in 50 µL TE.

## Expression of 1B capsid protein in Sf9, SDS-PAGE, and western blotting

To check the expression of 1B capsid protein in viral-transfected Sf9 cells, the released recombinant viruses in culture filtrates of Sf9 cells were collected and denoted as Stock-1 (P1). This stock was used for titration analysis via serial infection of Sf9 cells to propagate recombinant virions and eliminate non-recombinant virions. Additional two stocks, namely, Stocks-2 (P2) and -3 (P3), were obtained during this process. The recombinant virus from Stock-3 was used to infect Sf9 cells (10 PFU/cell) in 25-cm$^2$ tissue culture flasks (Greiner). Cells were allowed to grow for 72 h, harvested, and their total cellular proteins were analyzed on SDS-PAGE. In addition, the culture filtrates of infected Sf9 cells were also analyzed on SDS-PAGE to check for secretory 1B capsid protein. The expression of 1B capsid protein was evaluated by western blotting using monoclonal antibodies raised against 6×His-residues. In brief, cellular (C) and secretory (S) proteins were separated on 12% SDS-PAGE and the proteins were transferred onto a membrane (Immobilon® PVDF membrane; Millipore Corporation, Bedford, MA, United States) using a trans-blot apparatus (Bio-Rad), as described by *Towbin, Staehelin & Gordon (1979)*. The protein gel was incubated in the transfer buffer for 20 min, and the membrane was activated by soaking in ethanol for a few seconds. The membrane was washed with transfer buffer for 20 min and the protein was transferred to the membrane using a vertical mini-blotter at 25 V overnight. After the transfer was complete, the membrane was removed and blocked in TBS (50 mM Tris-Cl [pH 7.5] and 150 mM NaCl) containing 5% BSA and 0.05% Tween-20 for 1 h. The blocking buffer was discarded and replaced with TBS containing 10 µL of anti-His tag antibodies, and the membrane was incubated overnight at RT. The membrane was washed thrice with TBST, 5 min each, and incubated in TBS containing 4 µL of anti-mouse universal antibodies and kept overnight at RT. The membrane was washed thrice with TBST, as described above. Freshly prepared NBT/BCIP in alkaline phosphate buffer was added to the membrane and left at RT with agitation until the appearance of the purple band(s), indicating the locations of the specific protein(s).

## Indirect antigen-trapped ELISA

The sensitivity of recombinant capsid protein r1B to capture the FMDV antibodies in the serum of infected animals was assessed by indirect ELISA. Different concentrations of culture filtrate containing the r1B were coated in ELISA plate (Nunc; IL, United States) diluted in carbonate and bicarbonate buffer (pH 9.6). Total proteins concentrations per well were ranged from 400 to 12.5 ng and seeded in ELISA plates using 100 µL/well in duplicate (4 times) for each concentration. After overnight incubation at 4 °C, all wells

were washed using PBST (137 mM NaCl, 2.7 mM KCl, 8 mM Na2HPO4, 2 mM KH2PO4 and 0.1% Tween-20). Next, blocking buffer (PBS, pH 7.4 plus 5% non-fat milk and 3% BSA) was added (100 μL/well) to block the free residual spaces. Subsequently, the plates were incubated for 2 h at RT. The excess of the blocking buffer was removed, and the plates were washed twice using PBST. The positive sera from naïve calves infected with SAT 2 serotype, were diluted in PBS containing 3% BSA and incubated for 2 h at RT. After intensive washing, HRP-conjugated goat anti-bovine IgG (KPL; Kirkegaard & Perry Laboratories Inc., Gaithsburg, MD, United States) diluted 1:20000 in PBS containing 3% BSA was added (100 μL/well). After incubation at RT for 2 h, the plates were carefully washed with PBST and 100 μL of TMB-ELISA substrate (KPL) was added per well. To stop the reaction, 100 μL of stop solution (0.16 M sulfuric acid; ThermoFisher Scientific was added to each well. The plates were read at 450 nm, using a microplate reader (Vmax kinetic) and the absorbance value (OD) was determined as the cutoff point. The samples with OD $\geq$ 0.3 were considered positive, 0.2–0.3 as doubtful (should be repeated), and $\leq$ 0.2 as negative.

## RESULTS

### Cloning of 1B capsid gene in the pFastBac-Dual vector

The deduced amino acid sequence of 1B capsid protein of an Egyptian SAT-2 isolate (gb|AAZ83686) was used to determine the relatedness of several FMDV isolates, representing the six prevalent FMDV subtypes. Our data revealed that subtype-O was the most divergent when compared with other subtypes in our previous study (*Salem et al., 2019*). The 1B capsid protein was selected for the heterologous expression in insect cells and for antibody production, owing to its high sequence homology among several FMDV subtypes.

As illustrated in Fig. 1, the expression cassette contains the coding sequence of the 1B capsid protein, which was completely synthesized and its codon usage was altered to match better the preferred codon profile of *S. frugiperda* (the source for Sf9 cell line). Furthermore, the pFastBac-Dual cloning vector was modified to include the signal peptidase *sipS*-coding sequence under the control of the *p10* promoter to facilitate the secretion of the recombinant 1B capsid. The coding sequence of six His residues and an enterokinase recognition sequence were also introduced into the pFastBac-Dual vector and downstream from the 1B expression cassette. His-tag was used to monitor the expression of 1B capsid protein and used for further protein purification (Figs. 1 and 2A).

The successful cloning of the *SipS* gene into pFastBac Dual vector before and after cloning of the 1B expression cassette was verified. Endonucleases BbSI and NsiI were used to release the *SipS* fragment from the recombinant vector. As shown in Fig. 2B, the *SipS* gene fragment of 0.57 kb was successfully cloned and verified by restriction digestion, which confirmed the successful integration into the pFastBac-1B recombinant plasmid of 5.9 kb.

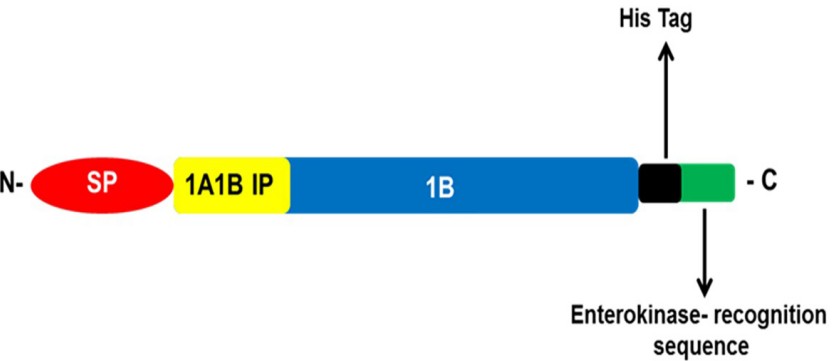

**Figure 1** The recombinant cassette, shows coding sequence of the 1A-1B inter-peptide (1A1B-IP is a short polypeptide linker that connects VP4/1A and VP2/1B, the signal peptide (SP), histidine tag, enterokinase recognition sequence and the 1B capsid protein, in fusion with a signal peptide.

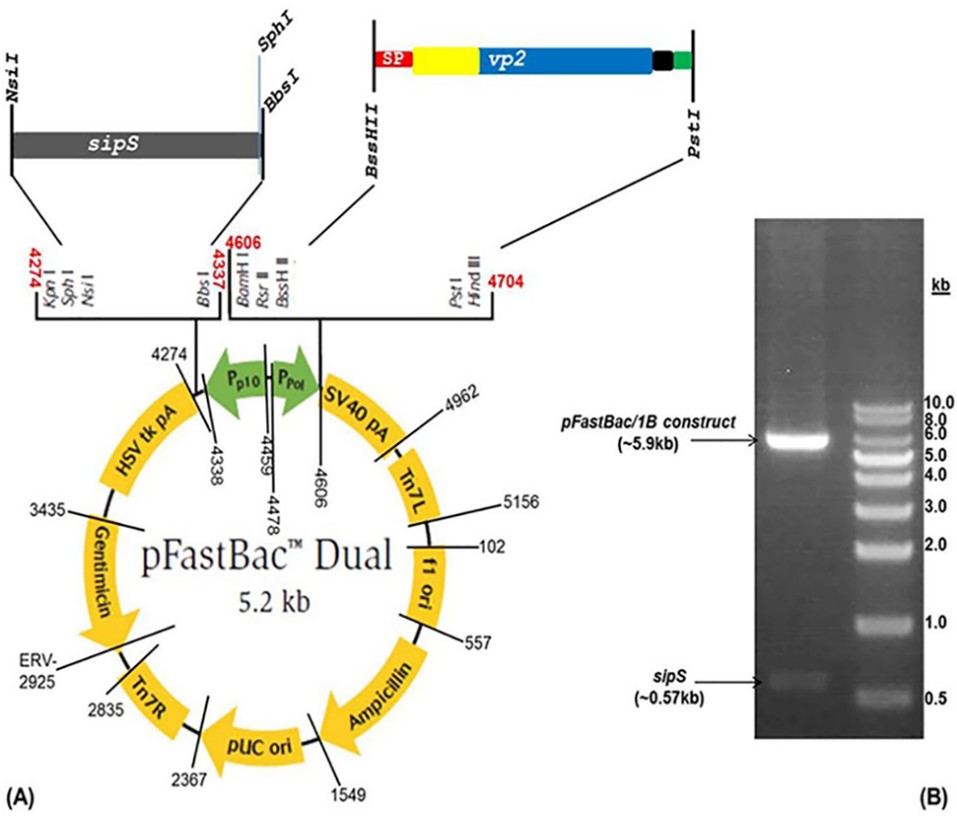

**Figure 2** Construction of pFastBacDual vector charboring the 1B cassette. (A) Plasmid map shows the recombinant cassette that was cloned downstream from the *Ppol* promoter of the modified pFastBac-Dual vector. (B) A total of 1% agaros gel shows the verified recombinant plasmid using restriction enzyme digestion. M: 1 kb ladder. RP: Recombinant plasmid harboring the 1B cassette and the SipS gene. Arrows show the molecular size (kb) for both SipS gene (0.57 kb) and the pFastBac-1B vector (5.9 kb).

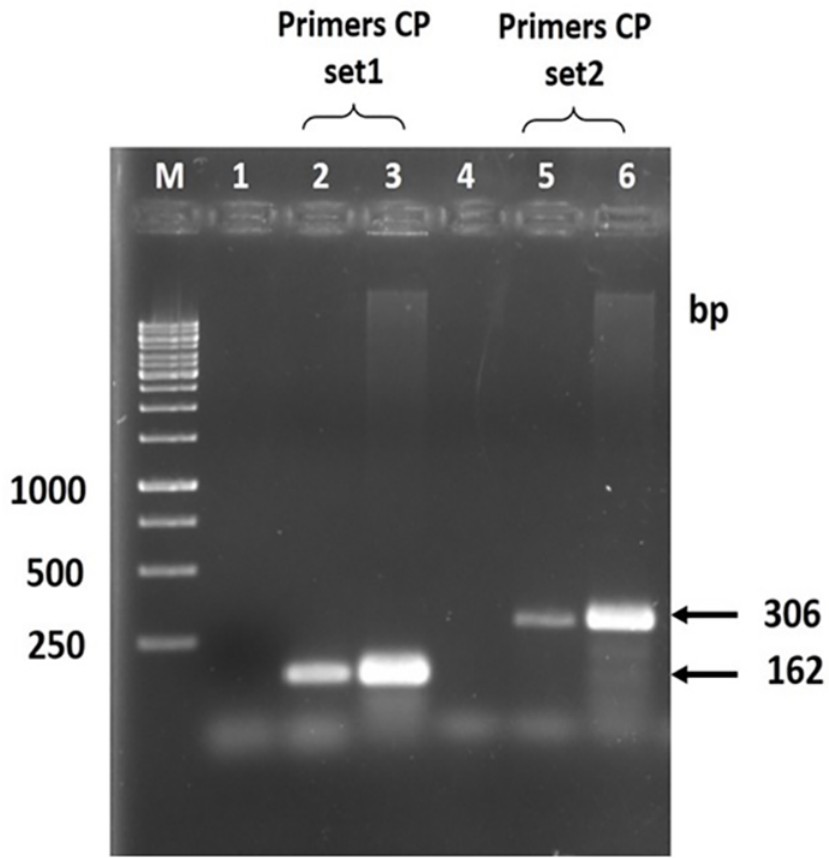

**Figure 3** **PCR detection of recombinant bacmid in infected S9 cells. M. 1kb Ladder, lane 1: Total DNA from Sf9 cells (mock infection), lane 2.** Total DNA from infected Sf9 cells, lane 3: pFastBacDual vector harboring the 1B cassette (positive control), lane 4: Total DNA from healthy Sf9 cells (mock infection), lane 5: Total DNA from infected Sf9 cells and lane 6: pFastBacDual vector harboring the 1B cassette (Positive control).

## Expression of 1B protein in Sf9 cells

To verify the transfection of recombinant bacmid into Sf9 cells, the presence of 1B coding sequence was detected by PCR. DNA extracted from transfected Sf9 cells was subjected to PCR using two pairs of primers. As shown in Fig. 3, PCR amplicons 162 bp and 306 bp, obtained by C1B-F/C1B-R1 and C1B-F/C1B-R2, respectively, were obtained. These results indicated the successful transfection of the recombinant bacmid into Sf9 cells.

The culture filtrate of infected Sf9 cells was analyzed using SDS-PAGE to check the expression of secretory 1B capsid protein. The expression of 1B capsid protein was evaluated by western blotting using subtype SAT 2-specific antibodies. As shown in Fig. 4A, a polypeptide of 22 kDa was detected in the cell lysate of viral infected Sf9 cells, but not in non-transfected cells. The identity of the 22-kDa polypeptide was confirmed by western blotting using SAT 2-specific polyclonal antisera. The results of the western blotting showed that the predicted molecular mass of 1B protein (22 kDa) was successfully
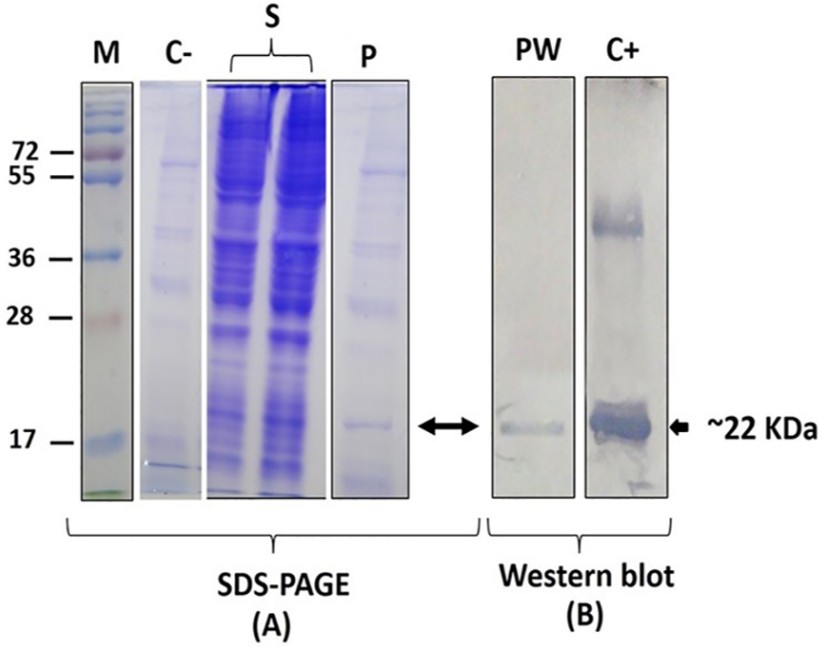

**Figure 4** **Protein analysis of 1B expressed in SF9 cells.** (A) SDS-PAGE (12%) of total protein extracted from culture filtrate of infected Sf9 cells using recombinant virus. M: Page Ruler prestained protein ladder. C-: Cells lysate extracted from mock infected Sf9 cells. S: Total protein of cells lysate extracted from viral infected Sf9 cells. P: Total protein extracted from culture filtrate (media) of infected Sf9 cells. (B) Western blot analysis of the expressed 1B capsid protein in Sf9 cells. PW: Secreted proteins in culture filtrate of Sf9 infected cells, C+: Positive control. Arrows show a clear protein band at ~22 kDa corresponding to 1B capsid protein in both SDS-PAGE gels and western blot.

detected using FMDV SAT 2-specific antibodies, suggesting the correct molecular mass of 1B protein using the baculovirus expression system (Fig. 4B).

## Indirect ELISA test

Serial dilutions ranging from 1:1 to 1:32 of P1 viral stock were used for coating in indirect ELISA to capture the FMDV SAT 2-specific antibodies in collected antiserum sample that was previously validated in VSVRI as positive by VNT.

The highest ELISA signals were detected from the dilution 1:1 of media containing the recombinant 1B protein. The OD values gradually decreased from the dilution 1:2 to dilution 1:16; no positive reaction was noticed with dilution 1:32 (Fig. 5).

## DISCUSSION

The primary aim of the present study was to express the 1B capsid polypeptide of FMDV in a soluble, less aggregated form, and evaluate its ability to detect the antibodies from FMDV-infected or -vaccinated animals. It was crucial to obtain the r1B capsid protein in a form capable of eliciting an adequate immune response and producing highly specific antibodies in immunized mice. Prior studies have reported the immunogenic properties of FMDV 1B protein (*Voller, Indwell & Bartlett, 1976*). Some studies have suggested that, although

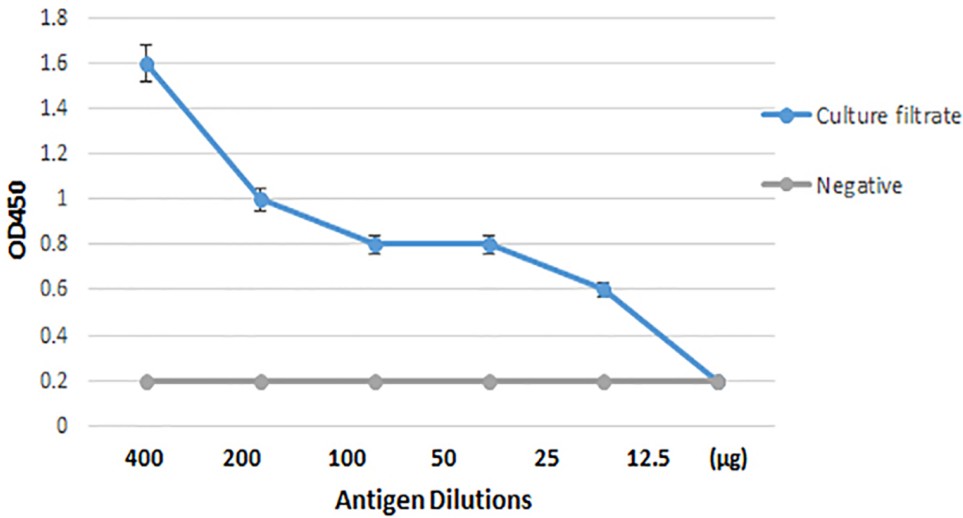

**Figure 5  Antigen sensitivity of r1B using indirect ELISA.** The sensitivity of the r1B capsid protein against FMDV SAT 2 specific antibodies was tested using ELISA assays. The $Y$-axis represents the optical density value (OD) measured at 450 nm. The $x$-axis represents serial dilutions of the Sf9-culture filtrate contain the r1B protein. The error bar represents 5% more or less of the ELISA reading value.

exposed to the viral surface, the N terminal region of 1B has low antigenic potential (*Lea et al., 1994*; *Curry et al., 1997*). However, the N terminus of 1B has been proven antigenic since it has no fixed position (*Acharya et al., 1989*), and its structural flexibility exposes some of the internal domains of the capsid proteins to the surface, allowing them to become definite antigenic sites (*Freiberg et al., 2001*). Another report (*Acharya et al., 1989*) showed in a similar vein that amino acid substitutions in the 1B subunit result in a significant antigenic diversity that leads to changes in FMDV's pathogenicity and replication capacity. Thus, even if some of the 1B N-termini sites are not accessible by Ab, the remaining accessible sites seem adequate for ELISA detection (*Freiberg et al., 2001*).

To achieve this, the baculovirus/Sf9 cells (BacSf9) expression system was selected for expressing the recombinant 1B capsid protein. Reasons for using BacSF9 system for protein expression included (1) secretion into the growth medium; (2) production of less aggregated proteins with an increased probability of correct folding, and (3) post-translational modifications for better antigenicity and specificity. Previously, the baculovirus expression system was successfully used to express different viral proteins such as the tomato yellow leaf curl virus coat protein (TYLCV) (*El-Gaied, Salem & Elmenofy, 2017*), Garlic mite-borne filamentous virus (GarMbFV) (*Ardisson-Arau'jo et al., 2013*), and glycoprotein E (gE) of the Egyptian BoHV-1.1 Abu-Hammad strain (rBac/gE-AbuH) (*El-Kholy et al., 2013*; *El-Gaied et al., 2020*). Furthermore, recombinant proteins produced using this system were validated for their antigenicity. The recombinant proteins were successfully used to detect viruses from infected samples, especially when the virus was present in low titers and mixed infection in plant hosts. As the 1B capsid protein is the most highly conserved capsid subunit among several FMDV subtypes, it was selected for the heterologous expression and further analysis. A compatible signal peptide/signal

peptidase system that ensured complete processing and secretion of expressed proteins was designed. Based on a previous study (*Ailor et al., 1999*; *Doel, 2003*; *Cao et al., 2009*; *Li et al., 2011*; *Salem et al., 2018*), we selected the signal peptidase gene *sipS* from *Bacillus subtilis*. Accordingly, the pFastBac-Dual cloning vector was modified by inserting the *sipS*-coding sequence under the control of the p10 promoter. The *sipS* coding sequence was completely synthesized using the published sequence; however, its codons were altered to match better the preferred codon usage of *S. frugiperda*, the source for Sf9 cell line. Furthermore, a signal peptide (SP) compatible with the sipS peptidase was incorporated upstream of the 1B coding sequence (Fig. 1) to facilitate the processing and secretion of the recombinant protein. Typically, the SP was 15 to 25 amino acids in length and consisted of three distinct regions: a short positively charged residue followed by a stretch of hydrophobic residues and ending with a few small side-chain residues located at the −3 and −1 position from the peptidase cleavage site, usually aspartic acid. We followed these criteria when we designed our SP; however, the yield of secreted proteins using the SP/SipS signal peptidase was relatively low (Fig. 4). This was probably due to the incomplete processing of SP; the SP was not completely removed. Accordingly, the r1B remained attached to the membrane and was not completely secreted in the medium. Another possibility was the production of proteins comprising mature as well as the precursor polypeptides. However, the anti-His antibody detected a single polypeptide corresponding to the correct size of 1B capsid subunit in western blotting; no other protein bands were detected (Fig. 4B, lane PW). The purified r1B capsid was used as a coated antigen in ELISA. The results showed that IB capsid protein-based indirect ELISA was able to detect anti-FMDV SAT 2 antibodies in sera raised against virus particles in all diluted samples. No signals were detected in all negative samples isolated from mock infected Sf9 cells. The clarified extract from cell culture filtrate used for coating the ELISA plates did not show interference or loss in specificity, suggesting no need for further purification (Fig. 5).

## CONCLUSION

In this study, the insect developed 1B-based ELISA has great potential to adapt to a quick pen-side test that can be performed on-farm or at animal quarantines. The purified IB capsid protein represents a promising candidate for detecting FMDV-infected animals using the protein-based indirect ELISA as a coating antigen. In addition, the purified protein may be used for raising a high concentration of virus-IB capsid protein antisera to be subsequently used in indirect 1B capsid-ELISA as a companion diagnostic test for FMDV detection. This would reduce the use of inactivated FMDV antigen in diagnostic assays, which will also increase the biosafety and geographic acceptability of the assay reagents.

## ACKNOWLEDGEMENTS

The authors would like to thank Dr. Alaa El-Kholy for providing us with FMDV SAT 2 specific antibodies.

### Funding

This work is supported by the Academy of Scientific Research and Technology (ASRT), Egypt (Jesor, grant number #49). The funders had no role in study design, data collection and analysis, decision to publish, or preparation of the manuscript.

### Grant Disclosures

The following grant information was disclosed by the authors:
Academy of Scientific Research and Technology (ASRT):  49.

### Competing Interests

The authors declare that there are no competing interests.

### Author Contributions

- Wael Elmenofy, Ismail Mohamed, Lamiaa El-Gaied, Reda Salem, Gamal Osman and Mohamed Ibrahim conceived and designed the experiments, performed the experiments, analyzed the data, prepared figures and/or tables, authored or reviewed drafts of the paper, and approved the final draft.

### Data Availability

Data are available as a Supplementary File.

### Supplemental Information

Supplemental information for this article can be found online at http://dx.doi.org/10.7717/peerj.8946#supplemental-information.

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
