# Peer review of "Expression of 1B capsid protein of Foot-and-mouth disease virus (FMDV) using baculovirus expression system and its validation in detecting SAT 2- specific antisera"

_PeerJ, doi:10.7717/peerj.8946_

## Round 0.1 · original submission · Minor Revisions

Please take into consideration the reviewer's comments, and provide a revised manuscript and a detailed point-by-point rebuttal letter.

Reviewer 1 ·

Basic reporting

no comment

Experimental design

I have a comment for studying the cross-reactivity of the developed protein in insect cells with seven serotypes

Validity of the findings

no comment

Additional comments

This article has described the procedure of expression of the 1B capsid protein of foot-and-mouth disease virus (FMDV) using baculovirus expression system and its validation in detecting SAT 2- specific antisera. I think the paper is excellent and written well (some minor things should be revised) and includes detailed steps of expression of 1B protein of FMDV based on baulovirus system. I have only minor comments, which are listed below:
1. The authors should add an experiment to illustrate the cross-reactivity of the developed 1B capsid protein of FMDV with several sera raised against several serotypes.
2. In ELISA, please change the dilution (1:1 to 1:32) to be a protein content for each dilution (µg)
3. It would be nice if you have another photo for western blot because the current photo has an air bubble above 1B capsid protein.
4. The plates were readied at 450 nm; please correct the title of the Y-axis in figure 5 and its legend.
5. Please add the host of SAT 2-specific antisera inline 213.
6. In line 117, change Luria broth (LB) to Luria-Bertani broth medium
7. please improve the quality of figure 5.

·

Basic reporting

Clear and unambiguous, professional English used throughout.

Experimental design

Good design

Validity of the findings

All underlying data have been provided; they are robust, statistically sound, & controlled

Additional comments

In this manuscript, the authors describe the expression of 1B capsid protein of FMDV using baculovirus expression system and validation of its antigenicity against SAT2 antibody. Toward this end, the authors constructed plasmid, performed transfection, infection, WB and ELISA assays. The results demonstrate the expression and specific reaction of the 1B capsid protein to SAT2 antibody, suggesting the expressed 1B capsid protein could be used for detection of FMDV due to infection or vaccination.
Overall, the manuscript is well written and the design of this study is straightforward. The results are encouraging and would be interesting for readers who work in FMD field.
Specific comments & questions:
1) After transfection, the author examined the presence of viral DNA in infected Sf9 cells by PCR. However, did they verify that by sequencing of the PCR product? If they did, describe it briefly.
2) How many passages did the authors conduct after transfection to propagate the recombinant baculovirus in Sf9 cells. Usually, a couple of passages would be need for a purpose of generating sufficient recombinant viruses.
3) Did the authors purify the 1B capsid protein expressed? It looked like the author purified the 1B protein, but no descriptions on purification in the manuscript. The purification procedures are suggested to be included.
4) In the M&M, the authors stated 100ul/well of the obtained r1B protein was used for iELISA, however, in the Result section, the authors stated a serial dilution of media ranging from 1:1 to 1:32 was used. What exact dilution was used? The volume of antigen does not make sense, an optimal quantity (weight, e.g. µg), rather thanvolume of the r1B protein, finally used for iELISA should be stated.
5) In the iELISA results shown in Fig 5, how many times or did the author repeat the expt? There was no error bar in the graph.
6) In the Discussion, more descriptions on the antigenicity of the expressed r1B are suggested to be expanded. In addition, further work, in particularly the potential application of the r1B protein in clinical detection should be described.

Reviewer 3 ·

Basic reporting

no comment

Experimental design

no comment

Validity of the findings

no comment

Additional comments

this manuscript is interesting and valuable, suitable for publication. the methodology is clear.
however, as a minor comment, I would prefer if the authors provide the sequence analysis showing the similarity of selected sequence of peptides in 1B capsid in SAT2 to the other FMDV subtypes. but the data provided is sufficient for publication .

---

## Round 0.2 · accepted · Accept

The manuscript has improved over the review rounds and it is now accepted at PeerJ.